# Dual-Task Optimization Method for Inverse Design of RGB Micro-LED Light Collimator

**DOI:** 10.3390/nano15030190

**Published:** 2025-01-25

**Authors:** Liming Chen, Zhuo Li, Purui Wang, Sihan Wu, Wen Li, Jiechen Wang, Yue Cao, Masood Mortazavi, Liang Peng, Pingfan Wu

**Affiliations:** Futurewei Technologies, 645 Martinsville Road, Basking Ridge, NJ 07920, USAjwang6@futurewei.com (J.W.);

**Keywords:** dual-task optimization, inverse design, micro-LED, incoherent light collimator, FDTD, light coupling efficiency

## Abstract

Miniaturized pixel sizes in near-eye digital displays lead to pixel emission patterns with large divergence angles, necessitating efficient beam collimation solutions to improve the light coupling efficiency. Traditional beam collimation optics, such as lenses and cavities, are wavelength-sensitive and cannot simultaneously collimate red (R), green (G), and blue (B) light. In this work, we employed inverse design optimization and finite-difference time-domain (FDTD) simulation techniques to design a collimator comprised of nano-sized photonic structures. To alleviate the challenges of the spatial incoherence nature of micro-LED emission light, we developed a strategy called dual-task optimization. Specifically, the method models light collimation as a dual task of color routing. By optimizing a color router, which routes incident light within a small angular range to different locations based on its spectrum, we simultaneously obtained a beam collimator, which can restrict the output of the light emitted from the routing destination with a small divergence angle. We further evaluated the collimation performance for spatially incoherent RGB micro-LED light in an FDTD using a multiple-dipole simulation method, and the simulation results demonstrate that our designed collimator can increase the light coupling efficiency from approximately 30% to 60% within a divergence angle of ±20° for all R/G/B light under the spatially incoherent emission.

## 1. Introduction

In recent years, the increasing requirement of the immersive experiences in AR/VR/MR devices has led to a rising demand for near-eye displays, which require significantly higher pixel densities than other digital devices, including tablets and smartphones, especially given the limited eye relief in most AR/VR devices [1]. First, there are three types of near-eye displays: augmented reality (AR), virtual reality (VR), and mixed reality (MR). The Apple Vision Pro and Meta Quest 3 are MR displays, which use video passthrough with cameras [2]. Several light engines, such as LCDs, OLEDs, laser beam scans, and liquid crystal on silicon have been used for AR/VR displays [3]. The pixel pitch of micro-organic light-emitting diode (micro-OLED) displays, consisting of neighboring red (R), green (G), and blue (B) channels, has been reduced to several microns. Though reducing pixel pitch will benefit the performance of the displays, such as mitigating the screen-door effect [4], the miniaturized pixel sizes will result in increased divergence angles of the emitted light, which scarifies the light coupling efficiency into image-forming optics, such as pancake lenses [5] in VR and MR devices and waveguides [6] in AR devices, making the energy efficiency of the devices not ideal.

Various strategies have been proposed to mitigate the issues caused by the large divergence angles of a micro-pixel emission. One typical approach involves integrating microlenses [7,8,9] within the pixel array to reduce the emission angle. However, the collimation effect of conventional lenses is wavelength-sensitive, causing color separation and noticeable patterns in near-eye displays. Alternatively, nanophotonic devices, such as metasurfaces [10], have been designed for beam shaping [11,12] or light collimation [13,14]. Since most of these devices manipulate light propagation by phase shifts produced by particularly designed meta-atoms with different dimensions, they are often used for coherent light such as lasers and waveguides. However, the emission light from micro-LEDs is typically incoherent with an unstable phase profile, making the designing of the metasurfaces extremely challenging. Though some works [15] have taken incoherent light into account, such as examining the performance of the devices under incoherent situations, still the designing processes only considered coherent light.

Meanwhile, complex or free-form nanophotonic devices have attracted extensive interest in recent years, as they can offer a higher degree of freedom in design flexibility and thus have the potential to perform complex functionalities, such as optical chips I/O [16,17], mode splitters [18,19], and beam shaping and steering [20,21]. However, designing such devices is challenging since the devices could be too complicated to be modeled by analytical approaches. Fortunately, the gradient descent optimization method, or the inverse design method, has been developed owing to the rapid development of computation capability with the pace of artificial intelligence. Instead of directly analyzing the light propagation using foundational physical principles such as the Huygens equivalence principle [10] or the generalized Snell’s law [22], the inverse design method quantifies the performance of the designed devices consisting of an array of parameters by deriving a figure of merit (FOM) (or objective function) from numerical electromagnetic simulation results. Then, partial derivatives of the FOM with respect to the parameters can be calculated efficiently using numerical methods such as the adjoint variable method [23] or automatic differentiation [24]. Finally, the parameters can be optimized using gradient descent algorithms. The inverse design method has been demonstrated as promising in designing nanophotonic devices for applications requiring an independent response towards incident light with a broadband spectrum, various polarization states, and different incident angles. For example, a subwavelength RGB color router [25,26,27], which is a nanophotonic device that routes incident light to different positions based on its spectrum, was designed for complementary metal–oxide semiconductor (CMOS) image sensors.

Based on the principle of optical reciprocity, we observed that the light path of color routing is a reverse of RGB micro-LED light collimation in a wide sense. Therefore, once we design a color router that works for incident light within a small angular range, we are expected to simultaneously obtain a beam collimator that can restrict the output of the light emitted from the color routing destination within a small divergence angle. We call this design strategy dual-task optimization given its similarity in concepts with the *duality* in mathematical optimization theory. The idea of this strategy is visualized in Figure 1.

In this work, we aim to design a nanostructure that covers the entire pixel containing R/G/B channels, as shown in Figure 1b, to reduce the large divergence angle, as shown in Figure 1a. Using our dual-task optimization strategy, instead of directly optimizing the collimator, we optimized a color router (shown in Figure 1c) using five incident angles within the range of −15° to 15° with a step size of 7.5° for a typical setup of a micro-LED display panel using the inverse design method with the finite-difference time-domain (FDTD) simulation method. To evaluate the collimation performance under an incoherent emission, we placed the designed structure on top of the pixels and ran 20 separate simulations for each channel with a corresponding dipole source at different locations within each channel. And we superposed the electrical field intensities of the 20 simulations as the output of the incoherent source to evaluate collimation performance. The simulation results demonstrate that the nanostructure obtained from our design strategy can improve the light coupling efficiency from approximately 30% to 60% within a divergence angle of ±20°for all R/G/B channels.

## 2. Materials and Methods

Figure 2 shows an overview of our design and evaluation methods in this work. In the following subsections, we will elucidate the details and principles behind our method.

### 2.1. Design Parameters

We first segmented the nanostructure into an array of designed units with a size of 10 nm × 10 nm. The height of the nanostructure was designed to be 2 μm, and the width was 4.5 μm to match the pixel pitch, which will be further discussed in Section 2.2. As shown in Figure 2a, the designed unit associated with the index (i, j) is first parameterized by a variable $v_{ij}\;\in \;R^{1}$; then, this variable is transformed to a design parameter $p_{ij}\in \;[0,\;1]$, which is realized by(1)$$p_{ij}=\;0.5\;\times \;\tanh(v_{ij})+1,$$

Then, each design parameter $p_{ij}$ is projected to a local permittivity value $\varepsilon_{ij}\in \;[\varepsilon_{min},\;\varepsilon_{max}$] using a binary projection function as(2)$$\varepsilon_{ij}=\varepsilon_{min}+\left(\varepsilon_{max}-\varepsilon_{min}\right)\;\times \;\frac{\tanh(\beta \eta )+\tanh(\beta (p_{ij}-\eta ))}{\tanh(\beta \eta )+\tanh(\beta (1-\eta ))},$$
where $\varepsilon_{min}$ and $\varepsilon_{max}$ correspond to the permittivity of the two materials that compose the nanostructure. In this work, we used silicon dioxide (SiO_2_) and silicon nitride (SiNx).

The corresponding permittivity can be obtained from the information in [28,29,30] and is summarized in Table 1. An empirical parameter β is used in the binary projection function to control the steepness of binarization. A large β value will force the projected permittivity snapping to the projection boundaries, resulting in a binarized structure feasible for fabrication (i.e., no region in the optimized structure is associated with fictitious permittivity in middle of the projection range). However, this will reduce the precision and numerical stability when calculating the derivatives for the optimization. Therefore, we used β=3 during the designing process. After finishing the entire inverse design process, we reprojected the designed parameters again using β=100 and then binarized all the parameters using the median value ${(\varepsilon }_{min}+\varepsilon_{max})/2$ as the threshold to ensure the structures are strictly binarized; *η* = 0.5 is the middle point for the projection function.

### 2.2. Micro-LED Pixel Structure

We used a typical setup of micro-LED pixels in our simulations, as shown in Figure 3a. This setup was also used during the inverse design of the color router process to ensure duality, except that the light sources were different. Since we simplified our simulations in 2D, a periodic pixel array along the *x*-direction in the *x*-*z* plane was studied. The width of each unit cell of the pixel array (outlined in blue, green, and red colors in Figure 3a) was 4.5 μm. Each pixel consisted of one blue, one green, and one red channel arranged from left to right. And each channel was composed of a multilayer slab comprising a 200 nm thick p-doped gallium nitride (GaN) layer, a 200 nm thick active multilayered quantum well (MQW), and a 700 nm thick n-doped GaN layer, from the top to the bottom. The width of each channel was 1.05 μm, and the channels were equally separated by polyimide filler. For simplicity, we ignored the bandwidth of the micro-LED light and only considered a typical wavelength (i.e., 457 nm, 527 nm, or 611 nm) for the blue, green, and red light. All the materials were characterized by their complex refractive indices at the wavelengths of interest, as summarized in Table 1.

### 2.3. Figure of Merit

The figure of merit (FOM) used in this work was derived in a similar sense as the work in [25,26]. The major difference was that, instead of only considering a plane-wave source with a single incident angle (i.e., 0°), we used 5 different incident angles $\theta \;\in \left\{-15{}^{\circ},\;-7.5{}^{\circ},0{}^{\circ},7.5{}^{\circ},15{}^{\circ}\right\}$, as shown in Figure 3a, to ensure that the designed structure can work within a small divergence angle. We positioned three planar field monitors $M_{i},i\in \left\{R,\;G,\;B\right\}$ at the middle height of the MQW region of the R/G/B channels, located 300 nm below the designed nanostructure. The FOM was then calculated using the data extracted from the field monitors as(3)$$J=\sum_{\theta }\left\{\sum_{color=R,G,B}\left[\alpha \mathrm{OE}\left(i\right)\;-\;\left(1-\alpha \right)\sum_{j\neq i}\mathrm{OX}\left(j,i\right)\right]\;\right\},$$
where OE(*i*) is the optical efficiency of the color channel *i*, quantified by the electrical field components routed to the correct channels matching their wavelengths. And OX(*j, i*) is the optical crosstalk from the color channel j into the color channel i, representing the wrong routings among the three channels. In this work, OE(*i*) and OX(*j, i*) can be obtained from the data of the three planer field monitors. For each incident angle, we ran 3 simulations for three different wavelengths. Then, we could obtain a 3 × 3 matrix $C={\left[c_{i,j}\right]}_{3\times 3}$ where $c_{i,j}$ is defined as(4)$$c_{i,j}=\int_{M_{i}}{\left|E_{x}^{j}\right|}^{2}dS$$
where $E_{x}^{j}$ is the *x*-component of the electric field at the wavelength corresponding to color *j*. Then, the diagonal terms of matrix C correspond to the OE(*i*), and the off-diagonal terms represent the crosstalk OX(*j, i*).

### 2.4. Collimation Evaluation Using Multiple-Dipole Method

We evaluated the performance of the incoherent light collimation after the designing process. Since there is no well-defined incoherent micro-LED source in the FDTD solver, we used a multiple-dipole method to better simulate the emission pattern of the micro-LEDs. Specifically, as shown in Figure 2b, we ran multiple separate simulations for a single channel. In each separate simulation, we placed the designed structure on top of a pixel and a single dipole source with the corresponding wavelength at a unique position within the corresponding channel. Considering that display panels contain periodic pixels, we duplicated multiple pixels and the designed structures along the *x*-direction, but only the middle blue, green, and red channels were excited by dipole sources. As shown in Figure 4a, in this work, we duplicated 5 pixels (i.e., in total, 15 R/G/B channels), as we found that further pixels will not have an obvious influence on the simulation results. After finishing all the separate simulations, we accumulated the angular power distributions in the far field, which can be easily obtained from a far-field projection monitor as the far-field angular power distribution under incoherent light. By repeating the same procedures for all R/G/B channels, we can evaluate the collimation performance by analyzing the power ratio within a small divergence angle and compare it with the results without the designed structure.

## 3. Results

In this work, we used a commercialized FDTD solver, *Tidy3D,* from Flexcompute Inc., to conduct all the simulations in the designing and evaluation processes. In order to reduce the computational cost, only two-dimensional (2D) simulations in the *x*-*z* plane were conducted in this work.

### 3.1. Results of Designing in Dual Task

We first built simulation environments in the FDTD to finish the designing of the structure in the dual task (i.e., color routing task). Figure 3a shows the simulation environments for a single-wavelength plane-wave source with five incident angles (i.e., −15°, −7.5°, 0°,7.5°,15°), and the zoom-in view shown on the right side demonstrates the schematic of the pixel structure as discussed in Section 2.2. Without loss of generality, the transverse magnetic (TM) polarization (*E_x_*, *H_y_*, *E_z_*) was adopted for the plane-wave sources. The Bloch boundary condition was adopted for the simulation boundaries in the *x*-direction to accommodate the plane-wave sources propagating at incident angles. Meanwhile, we set the boundary conditions along the *y*-direction as the periodic boundary conditions and the *z*-direction as the perfect matched layers (PML). We then used the adjoint variable method, which has been integrated in Tidy3D, to calculate the partial derivatives of our FOM in Equation (1) with respect to each of the design parameters discussed in Section 2.1. The adjoint variable method only requires two simulations to find all the partial derivatives, significantly speeding up the optimization process. With the partial derivatives, we used the Adam algorithm [31], which is a first-order gradient-based algorithm, to iteratively optimize the structure. In this work, we set the learning rate of the Adam algorithm as 0.3 and ran 15 iterations.

Figure 3b shows the optimization progress of the FOM values. After 15 iterations, we obtained the binarized structure following the method discussed in Section 2.1, as shown as the inset of Figure 3b. As discussed in Section 2.1, the structure is pixelated into an array of 10 nm × 10 nm element cells. Each cell shares the same geometry, but will be configured using two different materials, depending on the optimized parameters after the inverse design process. In this work, the final optimized parameters were transformed to two permittivity values using Equations (1) and (2), representing SiO_2_ and SiN_x_, respectively. We then imported the designed structure and used the same pixel geometries and materials, boundary conditions, and plane-wave sources in the FDTD to verify the success of the design. Figure 3c shows an example of the simulated results of the electric field profile when the incident angle is 15°. These results demonstrate that most of the light energy was successfully routed towards the desired channels. We also tested other incident angles, and the results are summarized in the histogram shown in Figure 3d. The results demonstrate that approximately 80% of the incident energy was correctly routed into the desired channels.

### 3.2. Results of Collimation Evaluation

Finally, we built another series of simulation environments, as discussed in Section 2.4., to evaluate the collimation performance, as shown in Figure 4a. Similar to the designing process, we ran separate simulations for R/G/B light. For each wavelength, we ran 20 simulations with only a single dipole source of the corresponding wavelength at a unique different location within the corresponding pixel. Then, the final electric field intensity of the incoherent micro-LED source can be viewed as the integration of the intensity obtained from the multiple dipole sources. After running the 20 simulations for the B, G, and R channels, respectively, we accumulated the electric field intensities (see Appendix A), which were calculated by the square of the electric field modulations, to obtain the intensity profiles, as shown in Figure 4b,d,f. We also calculated their corresponding far-field angular power ratio density distribution using the same method for the data obtained from a far-field projection angle monitor. The power ratio density was defined as(5)$$\bar{P_{r}}(\theta )=\;\frac{P(\theta )}{\sum_{\theta =-90{}^{\circ}}^{90{}^{\circ}}P(\theta )\times ∆\theta },$$
where ∆θ is the angle resolution and can be calculated by 180°/N, where N is the total number of projection angles. For comparison, we also ran the same number of simulations without the designed collimator to obtain the angular power distribution of the original micro-LEDs. The data and comparison shown in Figure 4c,e,g clearly demonstrate that our designed collimator successfully concentrates more power within a smaller divergence angle. We calculated the far-field power ratios within ±20° to numerically evaluate the performance of our designed structure. By dividing the summation of the power within ±20° by the total power on the far field, we found out that the power ratios were increased from 30.29%, 34.84%, and 35.83% to 65.60%, 55.94%, and 57.70% for the B, G, and R channels, respectively. Furthermore, we also evaluated the total energy efficiency. Specifically, we added a rectangular flux monitor surrounding the MQW region where the dipole sources will be placed, and we also added a planar flux monitor at a near-field position on top of the designed structure (i.e., the position of the far-field projection angle monitor) to calculate the percentage of the output flux. Our simulation results demonstrate that, in total, 27.82%, 24.84%, and 27.60% of the dipole source energy can be output, where the original percentages (i.e., without the collimator) were 26.63%, 26.53%, and 23.98%. This indicates that the designed structure did not cause notable energy loss. In summary, these simulation results demonstrate that the designed collimator can increase the light coupling efficiency from approximately 30% to 60% within a divergence angle of ±20° for all red, green, and blue channels under the spatial incoherent emission, and the designed structure will not cause a significant loss of total energy.

## 4. Discussion

We developed a designing strategy called dual-task optimization for the inverse design of a micro-LED light collimator. In this strategy, instead of directly optimizing the light collimation task, we found a dual task, which is the color routing, based on the principle of optical reciprocity. As a result, we obtained the collimator by optimizing the color routing task, and we evaluated that it achieved a satisfactory collimation performance. We validated that our method is an alternative approach for optimizing the collimator, and the major advantage is that it can work for an incoherent source, which is a challenging problem for direct optimization, as there is no well-defined incoherent source in the FDTD solver.

It is worth noting that the light path of the color routing is not a strict inverse of the collimation, since the wavefront of the dipole sources was not in a same form of the routed color. However, our simulation results demonstrate that even though the optical reciprocity is only in a wide sense, this idea can still work for the optimization.

The geometry (material distribution) of the fully optimized collimator is shown in the inset of Figure 3b. Some convex structures made of SiNx can be noticed within the collimator, which resembles the function of a conventional collimating lens, concentrating the LED emissions within a 20-degree divergence cone. However, the curvature of these SiNx convex structures, as well as their spatial arrangement, shows an irregular distribution, which is essential in combining the R/G/B emission to the same divergence angles, even though the R/G/B subpixels were at different locations. This irregular, free-form design is essentially difficult to achieve by using other first-principle, deterministic methods, which also highlights the necessity of our inverse design methodology.

Though the previous design can achieve a good collimation performance, fabricating such a nanostructure is extremely challenging, as the smallest feature size was 10 nm, which almost reached the limit of the current most advanced semiconductor manufacturing node. Therefore, we ran the dual-task designing process again while introducing a constraint to the smallest feature size of the device. Specifically, using the tool provided by Tidy3D, we employed a conic filter with a radius *r* to the design variables $v_{ij}$ before projecting them to the design parameters $p_{ij}$ using Equation (1) after each iteration. We set the filter radius *r* = 50 nm, which will constrain the smallest feature size to approximately $r/\sqrt{3}\approx \;28.87\;\mathrm{nm}$, aligning closely with the mature 28 nm semiconductor manufacturing node. The designed structure after using the fabrication constraint and its comparison with the collimator without a constraint are shown in Appendix A). We also evaluated the collimation performance again using the same method as discussed in Section 2.4 and Section 3.2. Appendix A) show the far-field angular power distribution for blue, green, and red light, respectively. The far-field power ratios within ±20° were 59.47%, 54.00%, and 57.97%. And the total energy efficiencies were 28.09%, 22.90%, and 24.19%, respectively. Comparing these results with the results shown in Section 3.2, we found out that the structure with the fabrication constraint can still improve the collimation performance, and the performance was not significantly diminished compared with the structure without a constraint. This result demonstrates the flexibility of the design considering the fabrication feasibility. Additional fabrication constraints and methods, such as those discussed in [32], can be further explored in the future to optimize the topology of the structure for higher fabrication feasibility. On the other hand, the fabrication processing flow for such a complex nanostructure is still a remaining challenge. Fortunately, the nano-fabrication technology is also in rapid development, and some work has been conducted in recent years to fabricate the nanophotonic devices in a similar form using semiconductor manufacturing technologies such as photolithography [33], electron beam lithography, and plasma etching [18,21].

## 5. Conclusions

In this work, we developed a design strategy for the inverse design in a nanophotonic collimator for micro-LEDs called dual-task optimization. We observed that the color routing, which was used in CMOS sensors, can be a dual task of a micro-LED light collimator. Then, by optimizing a color router, we simultaneously obtained a collimator. Finally, we validated the effectiveness of our dual-task optimization method through evaluating the performance of the collimator under a spatially incoherent situation, in which we ran separate emission simulations from 20 dipoles at various spatial locations in each R, G, and B channels and superposed the far-field power distribution together. The simulation results demonstrate that the designed collimator can successfully concentrate more power within a small divergence angle. Therefore, the designed collimator is promising for use in AR/VR devices such as near-eye displays to enhance the optical efficiency and thus reduce the power consumption.

## 6. Patents

The authors Z.L., P.W. and L.P. are listed as inventors on a patent application submitted to The Patent Cooperation Treaty (PCT) discussing the design and working principle of the designed nanostructure reported in this manuscript.

## Figures and Tables

**Figure 1 nanomaterials-15-00190-f001:**
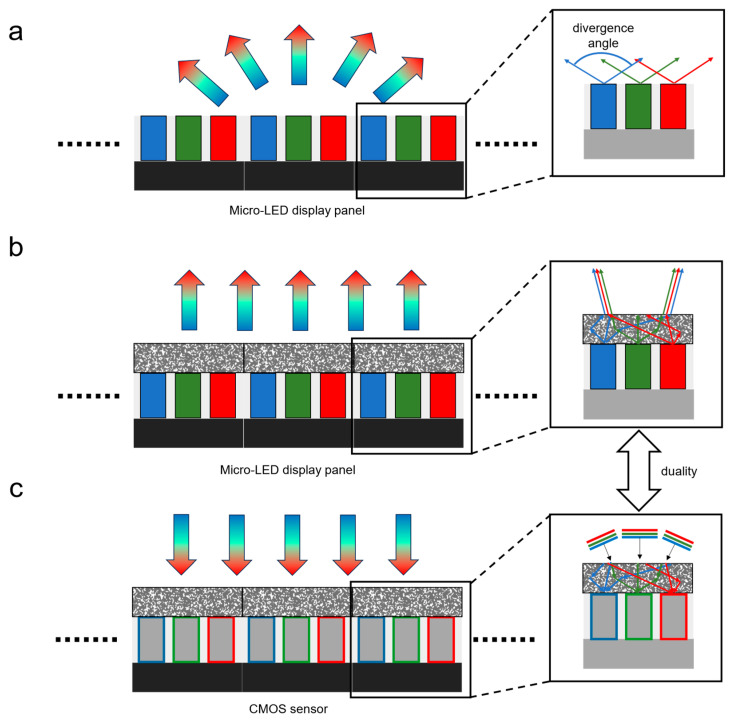
Conceptual schematic of the dual-design strategy for a beam collimator. (**a**) The original light emission of a typical micro-LED display panel. The divergence angle is very large. (**b**) Our solution of reducing the light divergence angle, i.e., designing a nanophotonic structure on top of each R/G/B channel. (**c**) The color router design of our dual-task optimization method. Instead of directly designing a collimator as in (**b**), we design a color router and use it as a collimator.

**Figure 2 nanomaterials-15-00190-f002:**
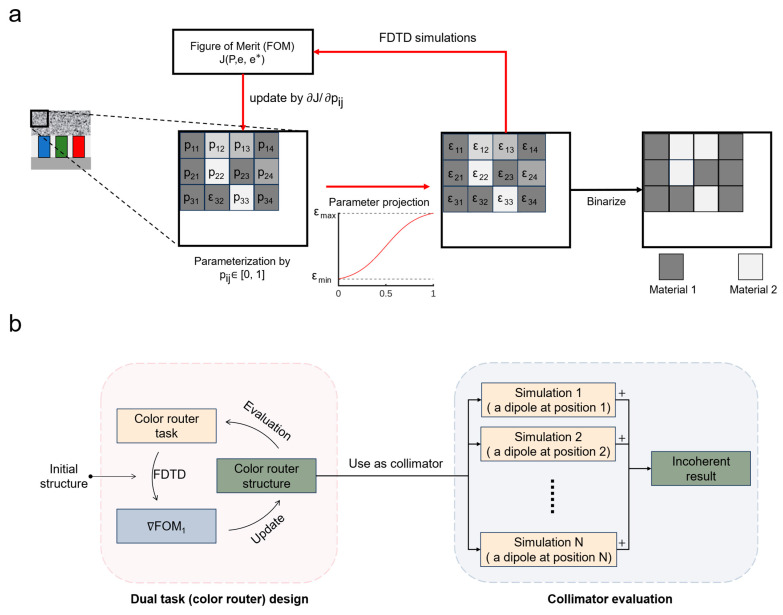
An overview of our dual-task optimization method for inverse design. (**a**) The design parameters of the collimator. (**b**) The procedures of our dual-task optimization for the collimator design.

**Figure 3 nanomaterials-15-00190-f003:**
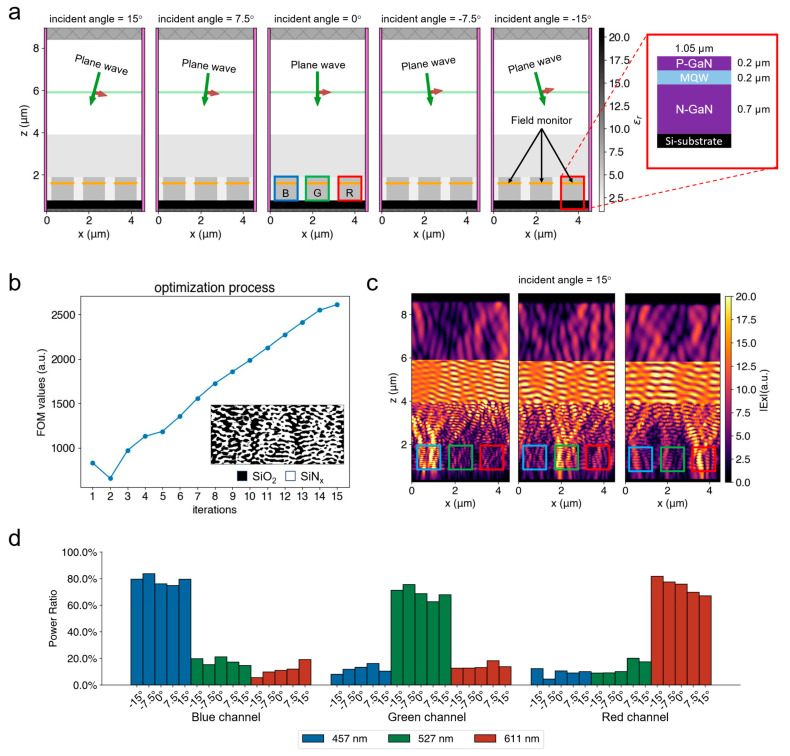
The simulation results of the designing in dual task. (**a**) The FDTD simulation environments used for the optimization process. In the environments, the blue arrows show the wave propagating directions and the red arrows show the polarization directions. (**b**) The optimization progress during the 15 iterations. The lower right corner shows the final binarized structure. (**c**) The modulation of the electrical fields for B/G/R light incident at 15° after using the designed structure. (**d**) Histogram showing the power ratio routed to the B/G/R channels after using the designed structure. The results demonstrate that approximately 80% of the light was correctly routed.

**Figure 4 nanomaterials-15-00190-f004:**
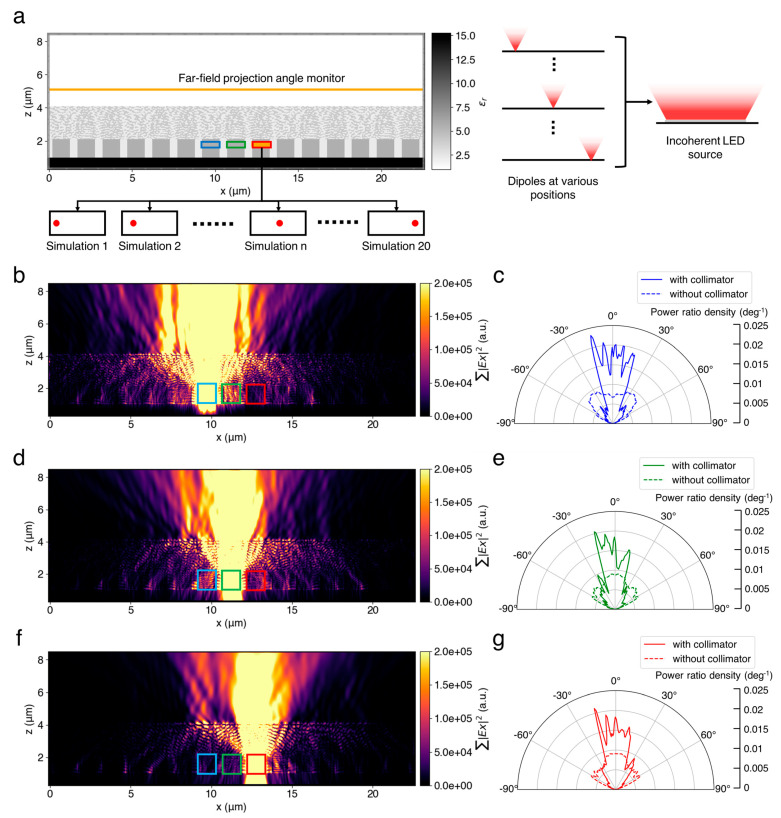
The simulation results for collimation performance evaluation. (**a**) The FDTD simulation environments. For each B/G/R channel, in total, 20 simulations were run, and each of them contain a dipole source at a different location. The accumulated results are considered as results with an incoherent micro-LED light source. (**b**,**c**) The accumulated electric field intensity profile and far-field angular power distribution for blue light. (**d**,**e**) The accumulated electric field intensity profile and far-field angular power distribution for green light. (**f**,**g**) The accumulated electric field intensity profile and far-field angular power distribution for green light. The original far-field angular power distributions of micro-LEDs without collimator are also plotted in (**c**,**e**,**g**) for comparison.

**Table 1 nanomaterials-15-00190-t001:** Summary of refractive indices of involved materials. The *n* and *k* represent the real and imaginary parts of the refractive indices of the involved materials, respectively. Data were acquired from references [28,29,30]. Reproduced from [29], with permission from Elsevier,1985.

Materials	Blue (457 nm)		Green (527 nm)		Red (611 nm)	
	*n*	*k*	*n*	*k*	*n*	*k*
GaN	2.49	\	2.38	\	2.35	\
MQW	2.51	\	2.41	\	2.37	\
Polyimide	1.5	\	1.5	\	1.5	\
SiO_2_	1.465	\	1.4609	\	1.4577	\
SiN_x_	2.0475	\	2.0279	\	2.0133	\
Silicon	4.583	0.13	4.215	0.048	3.906	0.022

## Data Availability

The data presented in this study are available on request from the corresponding author due to privacy.

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
