# Peer review of "Dual-Task Optimization Method for Inverse Design of RGB Micro-LED Light Collimator"

_nanomaterials, 2025, doi:10.3390/nano15030190_

Round 1
Reviewer 1 Report
Comments and Suggestions for Authors
This paper reports a theoretical study on a light collimator comprised of nanophotonic structures for micro-LEDs. The results may provide useful information for future design of micro-LED displays for AR smart glasses. I feel the paper is publishable in nanomaterials after a few revisions.
1) The details of the nanophotonic structure should be given. Is it a periodic array of SiO2 and SiNx? From Fig.3(b), it looks like a random mixture of SiO2 and SiNx.
2) The physics behind the collimating effect of the nanophotonic structure should be discussed briefly.
3) The authors assumed GaN micro-LEDs with a lateral dimension of approximately 1mm. In micron-sized micro-LEDs, most of the light is emitting into lateral direction through the sidewall surface. Is the collimation effect applicable to high-angle emission?
4) From Fig.1, the nanophotonic structure is directly placed on top of the GaN micro-LEDs. If SiO2 and SiNx are used as the materials for nanophotonic structures, there will be strong total internal reflection between the nanophotonic structure and the micro-LED. This will greatly reduce the coupling efficiency of light from micro-LEDs to the nanophotonic structure. Have the authors considered the influence of total internal reflection at the interface between micro-LEDs and the nanophotonic structure when they calculate the light coupling efficiency of the collimator?
Author Response
This paper reports a theoretical study on a light collimator comprised of nanophotonic structures for micro-LEDs. The results may provide useful information for future design of micro-LED displays for AR smart glasses. I feel the paper is publishable in nanomaterials after a few revisions.
Response: Thank you for taking your time reviewing our manuscript. We are glad that you like our work.
Comments 1: The details of the nanophotonic structure should be given. Is it a periodic array of SiO2 and SiNx? From Fig.3(b), it looks like a random mixture of SiO2 and SiNx.
Response 1: Thanks for pointing this out. The designed nanostructure is indeed a mixture of SiO2 and SiNx. However, they are neither periodic nor randomly mixed. The whole structure is an array of units with a size of 10 nm × 10 nm. Each unit can be either SiO2 or SiNx, depending on the optimized permittivity distribution. And the permittivity distribution is inverse designed, i.e., solved by gradient descent algorithms, instead of manually configured. We revised the second paragraph of Section 3.1 (starting from Line 203)
“As discussed in Section 2.1, the structure is pixelated into an array of 10 nm × 10 nm element cells. Each cell shares the same geometry, but will be configured using two different materials, depending on the optimized parameters after the inverse design process. In this work, the final optimized parameters were transformed to two permittivity values using Equation 1 and Equation 2, representing SiO2 and SiNx, respectively.”
We hope the revised content can help readers understand the idea better.
Comments 2: The physics behind the collimating effect of the nanophotonic structure should be discussed briefly.
Response 2: Thanks for the advice. On the Discussion section (i.e., Section 4), we added the following paragraph on Line 266 to further discuss the physics of the nanophotonic structure.
“The geometry (material distribution) of the fully optimized collimator is shown in the inset of Figure 3(b). Some convex structures made of SiNx can be noticed within the collimator, which resembles the function of conventional collimating lens, concentrating the LED emissions within 20-degree divergence cone. However, the curvature of these SiNx convex structures, as well as their spatial arrangement, shows an irregular distribution, which are essential to combine the R/G/B emission to the same divergence angles, even though R/G/B subpixels were at different locations. This irregular, free-form design is essentially difficult to achieve by using other first-principle, deterministic method, which also highlights the necessity of our inverse design methodology.”
We hope the revised content can help readers understand the idea better.
Comments 3: The authors assumed GaN micro-LEDs with a lateral dimension of approximately 1mm. In micron-sized micro-LEDs, most of the light is emitting into lateral direction through the sidewall surface. Is the collimation effect applicable to high-angle emission?
Response 3: The lateral dimension of a micro-LED subpixel in our work is 1.05 µm. Therefore, our simulation results have demonstrated that the collimator can work for micon-sized micro-LEDs. Also, the light emitting into the lateral direction through the sidewall surface has also been taken into account during our optimization process since we used dipoles as the source, and the dipoles will emit electromagnetic wave within 360°. Therefore, the collimation effect is applicable to high-angle emission.
Comments 4: From Fig.1, the nanophotonic structure is directly placed on top of the GaN micro-LEDs. If SiO2 and SiNx are used as the materials for nanophotonic structures, there will be strong total internal reflection between the nanophotonic structure and the micro-LED. This will greatly reduce the coupling efficiency of light from micro-LEDs to the nanophotonic structure. Have the authors considered the influence of total internal reflection at the interface between micro-LEDs and the nanophotonic structure when they calculate the light coupling efficiency of the collimator?
Response 4: This is a great question. In our work, we have actually considered the influence of total internal reflection at the interface between micro-LEDs and the nanophotonic structure when calculating the light coupling efficiency. Besides comparing the far-field energy ratio within the 20-degree emission angle, we also calculated the percentage of the total output energy, i.e., the light that was emitted out of the top of the nanophotonic structure in Line 245 – 251. Considering the simplicity of the optimization, though we did not include constraints of the total energy efficiency in the figure of merit, the results demonstrate that the total energy efficiency was almost kept at the similar level. Therefore, the total internal reflection did not affect the performance of the designed structure.

Reviewer 2 Report
Comments and Suggestions for Authors
Please see attached review report.

Author Response
In this manuscript, the authors developed a dual-task optimization for the inverse design of a nanoscale microLED light collimator. The metasurface is optimized using the Adam algorithm, resulting in an increase in light coupling efficiency from approximately 30% to 60% within a ±20° divergence angle. However, the fabrication of such nanostructures remains challenging. The introduction part requires significant improvement, and the modeling and optimization procedures are not sufficiently explained. Therefore, this work can be considered for publication after minor revisions.
Response: Thank you for taking your time reviewing our manuscript. We are glad that you like our work.
Comments 1: Introduction: For better clarity, please rewrite the first paragraph. First, there are three types of neareye displays: augmented reality (AR), virtual reality (VR), and mixed reality (MR). Apple Vision Pro and Meta Quest 3 are MR displays, which use video passthrough with cameras, as described in detail by Z. Yang, et al. “Advances and challenges in microdisplays and imaging optics for virtual reality and mixed reality,” Device 2 (2024) 100398. Second, line 36, pancake lenses are used in VR and MR, but not in AR. Third, several light engines, such as LCDs, OLEDs, Laser beam scans, Liquid-crystal-onsilicon, etc. for AR/VR displays have been reviewed by E. L. Hsiang, et al. “AR/VR light engines: perspectives and challenges,” Adv. Opt. Photon. 14(4) (2022) 783-861. These references will help readers to better understand the background information.
Response 1: Thank you for your suggestions. After checking the mentioned papers, we agree that these references will help readers to better understand the background. Therefore, we have rewrite the first paragraph and added these references.
Comments 2: Introduction, 2nd paragraph: In addition to the methods introduced by the authors, the following approach is also relevant: E. L. Hsiang, et al. “Tailoring the light distribution of micro-LED displays with a compact compound parabolic concentrator and an engineered diffusor,” Opt. Express 29(24) (2021) 39859-39873.
Response 2: Thank you for the suggestion. After carefully reading this paper, we have added this paper in our references 9.
Comments 3: Line 100 and Figure 2(a): For the 10 nm × 10 nm size, please specify whether this refers to the size of each unit or the size of each element in the array. In addition, please clarify if each element in the unit shares the same geometry and material properties, or if they are individually designed.
Response 3: The 10 nm × 10 nm size is indeed the size of each unit (i.e., each element in the array), as depicted in the zoom-in illustration in Fig. 2a. All the elements share the same geometry but can be configurated with different material properties (either SiO2 or SiNx). In the design progress, we used a binarized 2D array (Fig. 2a, rightmost panel) to represent a specific material configuration, which is designed through the dual-task optimization. To make things more clear, we revised the second paragraph of Section 3.1 (starting from Line 205) as follows:
“As discussed in Section 2.1, the structure is pixelated into an array of 10 nm × 10 nm element cells. Each cell shares the same geometry, but will be configured using two different materials, depending on the optimized parameters after the inverse design process. In this work, the final optimized parameters were transformed to two permittivity values using Equation 1 and Equation 2, representing SiO2 and SiNx, respectively.”
Comments 4: Equation 2: What does η stand for?
Response 4: The η is a parameter that can control the middle point of the binary projection function. In this work, we set this parameter as a constant of 0.5. We have added the explanation at the end of Section 2.1 (Line 123)
“η = 0.5 is the middle point for the projection function. ”
Comments 5: Line 110: Please illustrate how the empirical parameter β influences the geometry of the nanostructures.
Response 5: The parameter β controls the steepness of the binarization. A larger β will produce more binarized permittivity distribution. In order to help readers better understand this parameter, we created a new figure in supplementary materials as follows.
Figure S6. The effect of different β towards the binary projection function. The parameter η is set as 0.5.
Comments 6: Explanations in line 179 mention that the simulation is 2D, while the description in lines 188-191 indicates that the boundary condition of the simulation is 3D. Please clarify.
Response 6: We conducted the simulations in 2D by setting the dimension along the y-direction as 0. However, the boundary conditions for all the three directions (i.e., x, y, and z) are still required by the FDTD method. In this work, we set the boundary condition along the y-direction as the periodic boundary condition, which is a common practice when only concern the results along the x-z plane in FDTD simulations.
Comments 7: Please describe the collimator’s geometry and properties after optimization.
Response 7: Thanks very much for the wonderful question. To describe the collimator’s geometry more clearly, we revised the second paragraph of Section 3.1 (starting from Line 203). To describe the physics of this collimator, we added the following paragraph on Discussion section (i.e. Section 4, starting Line 266).
“The geometry (material distribution) of the fully optimized collimator is shown in the inset of Fig. 3b. Some convex structures made of SiNx can be noticed within the collimator, which resembles the function of conventional collimating lens, concentrating the LED emissions within 20-degree divergence cone. However, the curvature of these SiNx convex structures, as well as their spatial arrangement, shows an irregular distribution, which are essential to combine the R/G/B emission to the same divergence angles, even though R/G/B subpixels were at different locations. This irregular, free-form design is essentially difficult to achieve by using other first-principle, deterministic method, which also highlights the necessity of our inverse design methodology.”
We hope the revised content can help readers understand the idea better.

Round 2
Reviewer 1 Report
Comments and Suggestions for Authors
The authors have addressed all my comments properly. I think the paper is now ready for publication.